

# Probabilistic biomechanical finite element simulations: whole-model classical hypothesis testing based on upcrossing geometry

Todd C. Pataky[1,2], Michihiko Koseki[2] and Phillip G. Cox[3,4]

[1] Institute for Fiber Engineering, Shinshu University, Ueda, Japan
[2] Department of Bioengineering, Shinshu University, Ueda, Nagano, Japan
[3] Hull York Medical School, Centre for Anatomical and Human Sciences, University of York, York, UK
[4] Department of Archaeology, University of York, York, UK

Corresponding author
Todd C. Pataky,
tpataky@shinshu-u.ac.jp

## ABSTRACT

Statistical analyses of biomechanical finite element (FE) simulations are frequently conducted on scalar metrics extracted from anatomically homologous regions, like maximum von Mises stresses from demarcated bone areas. The advantages of this approach are numerical tabulability and statistical simplicity, but disadvantages include region demarcation subjectivity, spatial resolution reduction, and results interpretation complexity when attempting to mentally map tabulated results to original anatomy. This study proposes a method which abandons the two aforementioned advantages to overcome these three limitations. The method is inspired by parametric random field theory (RFT), but instead uses a non-parametric analogue to RFT which permits flexible model-wide statistical analyses through non-parametrically constructed probability densities regarding volumetric upcrossing geometry. We illustrate method fundamentals using basic 1D and 2D models, then use a public model of hip cartilage compression to highlight how the concepts can extend to practical biomechanical modeling. The ultimate whole-volume results are easy to interpret, and for constant model geometry the method is simple to implement. Moreover, our analyses demonstrate that the method can yield biomechanical insights which are difficult to infer from single simulations or tabulated multi-simulation results. Generalizability to non-constant geometry including subject-specific anatomy is discussed.

## INTRODUCTION

In numerical finite element (FE) simulations of biomechanical continua model inputs like material properties and load magnitude are often imprecisely known. This uncertainty arises from a variety of sources including: measurement inaccuracy, *in vivo* measurement inaccessibility, and natural between-subject material, anatomical and loading variability (*Cheung et al., 2005*; *Ross et al., 2005*; *Cox et al., 2011*; *Cox, Rinderknecht & Blanco, 2015*; *Fitton et al., 2012b*). Despite this uncertainty, an investigator must choose specific parameter

values because numerical simulation requires it. Parameters are typically derived from published data, empirical estimation, or mechanical intuition (*Kupczik et al., 2007*; *Cox et al., 2012*; *Cox, Kirkham & Herrel, 2013*; *Rayfield, 2011*; *Cuff, Bright & Rayfield, 2015*).

It is also possible to perform multiple FE simulations using a spectrum of feasible model input values to generate a distribution of model outputs (*Dar, Meakin & Aspden, 2002*; *Babuska & Silva, 2014*). More simply, probabilistic model inputs yield probabilistic outputs, and continuum mechanics' inherent nonlinearities ensure that these input and output probabilities are nonlinearly related. Probing output distributions statistically therefore generally requires numerical simulation. Such analyses can require substantial computational resources: probabilistic FE outputs have been shown to converge to stable numerical values only for on-the-order of 1000 to 100,000 simulation iterations depending on model complexity (*Dopico-González, New & Browne, 2009*). The advent of personal computing power has mitigated problems associated with this computational demand and has led to a sharp increase in probabilistic FE simulation in a variety of engineering fields (*Stefanou, 2009*) including biomechanics (*Easley et al., 2007*; *Laz et al., 2007*; *Lin et al., 2007*; *Radcliffe & Taylor, 2007*; *Fitzpatrick et al., 2012*).

Producing a probabilistic input–output mapping is conceptually simple: iteratively change input parameters according to a particular distribution and assemble output parameters for each iteration to yield an output distribution. The simplest method is Monte Carlo simulation which randomly generates input parameters based on given mean and standard deviation values (*Dar, Meakin & Aspden, 2002*). More complex methods like Markov Chain Monte Carlo can accelerate probabilistic output distribution convergence (*Boyaval, 2012*).

Once probabilistic inputs/outputs are generated they may be probed using a variety of statistical methods. A common technique is to extract scalars like maximum von Mises stress from anatomically demarcated regions of interest (*Radcliffe & Taylor, 2007*). Other techniques include Taguchi global model comparisons (*Taguchi, 1987*; *Dar, Meakin & Aspden, 2002*; *Lin et al., 2007*) to fuzzy set modeling (*Babuska & Silva, 2014*) and probability density construction for specific model parameters (*Easley et al., 2007*; *Laz et al., 2007*; *McFarland & Mahadevan, 2008*; *Dopico-González, New & Browne, 2009*).

The purpose of this paper is to propose an alternative method which conducts classical hypothesis testing at the whole-model level using continuum upcrossing geometry. An 'upcrossing' is a portion of the continuum that survives a threshold (Fig. 1) like an island above the water's surface or a mountain top above clouds. Each upcrossing possess a number of geometrical features including maximum height, extent and integral, where integrals, for examples, are areas, volumes and hyper-volumes for 1D, 2D and 3D continua, respectively. Parametric solutions to upcrossing geometry probabilities exist for *n*-dimensional Gaussian continua in the random field theory (RFT) literature (*Adler & Taylor, 2009*), and non-parametric approximations have been shown to be equally effective (*Nichols & Holmes, 2002*). The method we propose follows the latter, non-parametric permutation approach because it is ideally suited to the iterative simulation which characterizes probabilistic FE analysis.

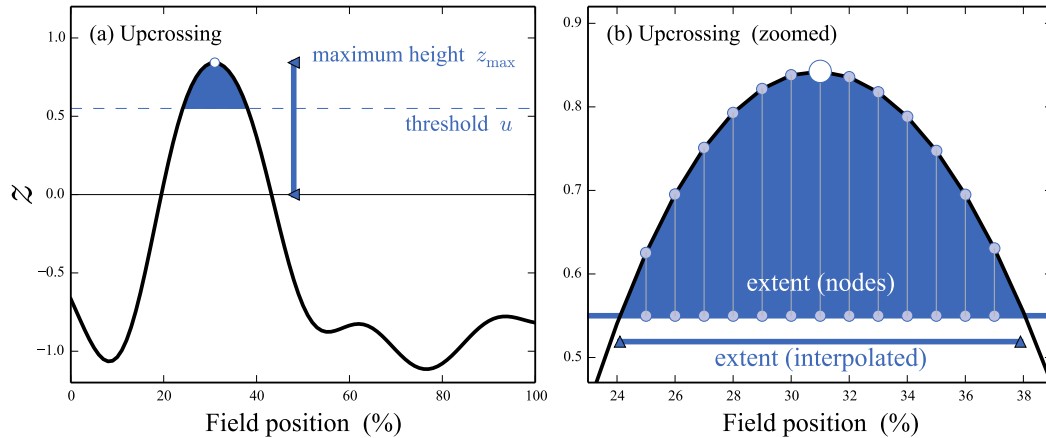

**Figure 1** **Example upcrossing in a 1D continuum.** A thresholded continuum contains zero or more upcrossings, each with particular geometric characteristics including: maximum height, extent, integral, etc., each of which is associated with a different probability. The maximum height characteristic—across all upcrossings—can be used to conduct classical hypothesis testing as described in 'Methods.'

The method is inspired by hypothesis testing approaches in nonlinear modeling (*Legay & Viswanatha, 2009*) and in particular a label-based continuum permutation approach (*Nichols & Holmes, 2002*). It first assembles a large number of element- or node-based test statistic volumes through iterative simulation, then conducts inference using non-parametrically estimated upcrossing probabilities. These upcrossing distributions form a general framework for conducting classical, continuum-level hypothesis testing on FE models in arbitrarily complex experiments.

## METHODS

All analyses were were implemented in FEBio v.2.4.2 and v.2.5.0 (*Maas et al., 2012*) and Python 2.7 (*Van Rossum, 2014*). All partial differential equations underlying the models' numerical solutions are described in the FEBio Theory Manual (*Maas et al., 2015*). Model files and analysis scripts are available in this project's GitHub repository (http://github.com/0todd0000/probFEApy).

### Models
#### *Model A: simple anisotropic bone compression*
A single column of hexahedral elements (Fig. 2A) with anisotropic stiffness (Fig. 2B) was used to represent bone with local material inconsistencies. This simplistic model was used primarily to efficiently demonstrate the key concepts underlying the proposed methodology. Nodal displacements were fully constrained at one end, and a total compressive force of 8,000 N was applied to the other end along the longitudinal axis. The bone material was linearly elastic with a Poisson's ratio of 0.3.

Local anisotropy in Young's modulus (Fig. 2B) was created using Gaussian pulses centered at 70% along the bone length with amplitudes and breadths of approximately 10% and 20%, respectively. The actual amplitudes and breadths of the stiffness increase were varied randomly to simulate an experiment involving $N = 8$ randomly sampled

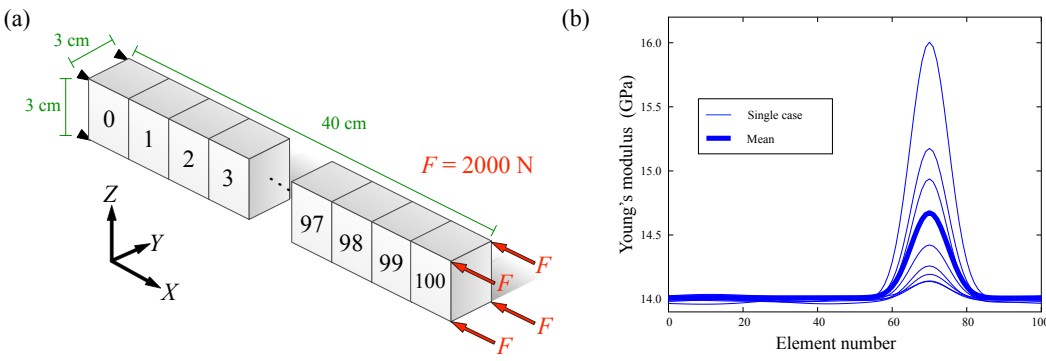

**Figure 2** **Model A.** (A) Stack of cuboids representing a simplified bone. (B) Elemental Young's moduli representing local stiffness increase in $N = 8$ cases.

subjects in which the bone's anisotropic stiffness profile was measured separately for each subject. Additionally, a small random signal was separately applied to each of the eight cases to ensure that variance was greater than zero, and thus that test statistic values were computable at all points in the continuum.

### Model B: soft tissue indentation

A rigid hexahedral block was compressed against soft tissue to a depth of 1 cm height as depicted in Fig. 3. Nodal displacements on the soft tissue's bottom surface were fully constrained. The soft tissue was modeled as hyperelastic with the following Moony–Rivlin strain energy function (*Maas et al., 2015*):

$$W = a(I - 3) + \frac{k}{2}(\ln J)^2. \tag{1}$$

Here $a$ is the hyperelastic parameter, $k$ is the elasticity volume modulus, $I$ is the deformation tensor's first deviatoric invariant, and $J$ is the deformation Jacobian. The parameter $a$ was set to 100 and eight $k$ values (800, 817, 834, 851, 869, 886, 903, 920) were compared to a datum case of $k = 820$.

Additionally, three different indenter face types were compared. The first indenter face was perfectly flat, and the other two were uneven but smooth as depicted in Fig. 4. The uneven surfaces were generated by adding spatially smoothed Gaussian noise to the indenter face's z coordinates (i.e., the compression direction), then scaling to a maximum value of approximately 2.5 mm, or 1.7% the indenter's height.

### Model C: hip cartilage compression

A separately-published model of hip cartilage compression (*Maas et al., 2015*) (Fig. 5) was selected to demonstrate how the concepts from the simple models A and B above may extend to realistic biomedical applications. This model is available in the FEBio test suite (http://febio.org; model name: "hip_n10rb"), and the scripts we used to manipulate this model are available in this paper's GitHub repository (http://github.com/0todd0000/probFEApy).

The bones were rigid and the cartilage was modeled using the hyperelastic Mooney-Rivlin model above Eq. (1) with a constant $a$ value of 6.817. Ten different values of $k$

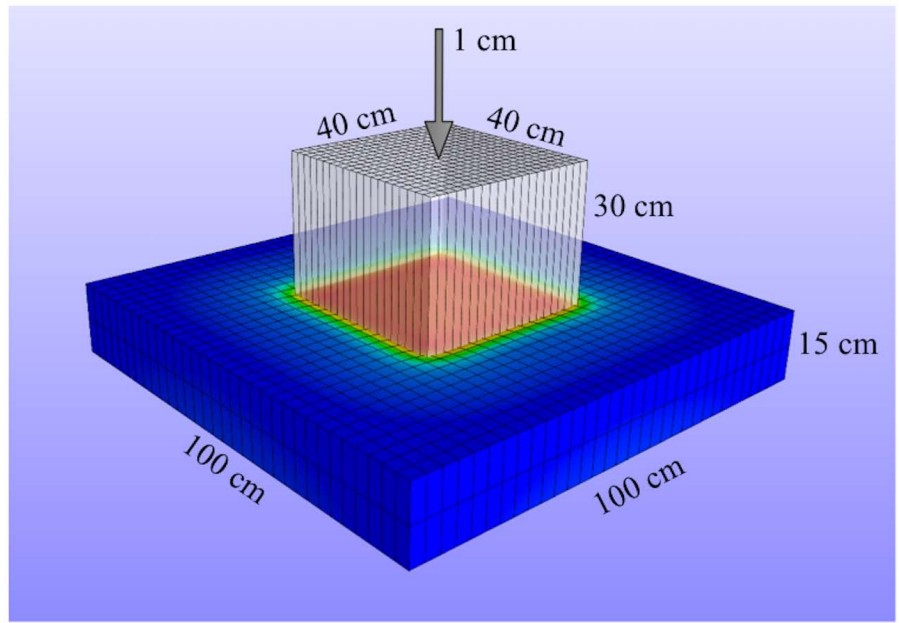

**Figure 3** **Model B.** Rigid block indentation on a hyperelastic material.

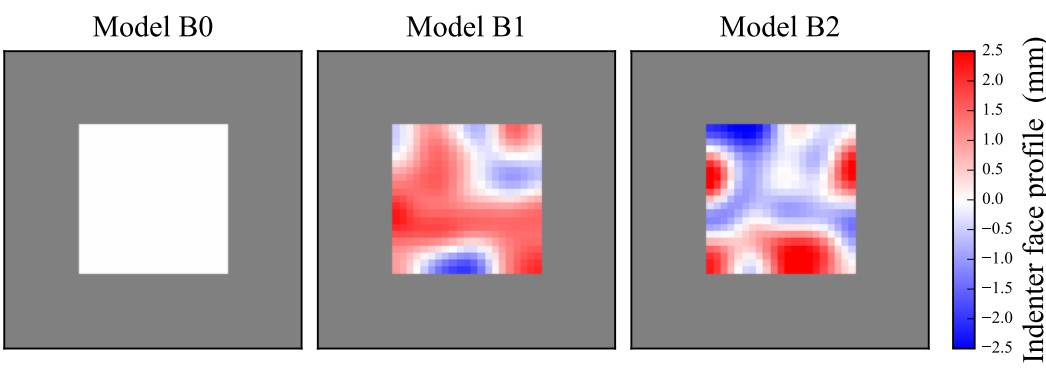

**Figure 4** **Model B indenter faces.** The grey area depicts the compressed soft tissue.

were simulated for each of two hypothetical groups (Table 1) to mimick a two-sample experiment involving *in vivo* or *in vitro* material property measurements. The pelvis and acetabular cartilage were fixed and the femur was kinematically driven 1 mm in the upward direction.

## Analysis

We used a non-parametric permutation method from the Neuroimaging literature (*Nichols & Holmes, 2002*) to conduct classical hypothesis testing at the whole-model level. The technique employs observation permutation to generate non-parametric approximations to probabilities from (parametric) multi-dimensional Random Field Theory (*Adler & Taylor, 2009*). The method is described below and is depicted in Fig. 6. All permutations described below were applied to pre-simulated FEA results.

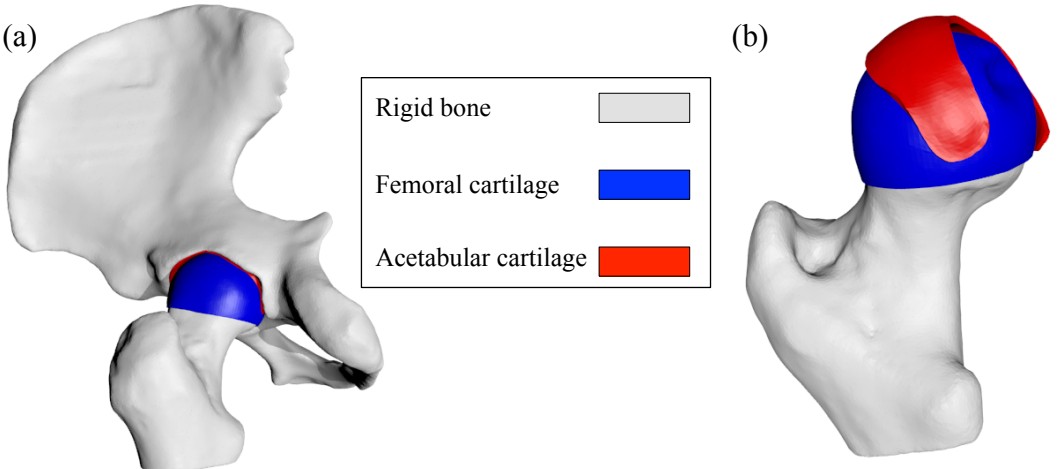

**Figure 5** **Model C.** "hip_n10rb" from the FEBio test suite containing femoral and acetabular cartilage compressed via rigid bone displacement. (A) Full model. (B) Pelvis removed to expose the cartillage surface geometries.

**Table 1** **Model C material parameters.** See Eq. (1). SD, standard deviation.

| Group | Mooney–Rivlin $k$ values | Mean (SD) |
|---|---|---|
| 1 | [1,200, 1,230, 1,260, 1,290, 1,320, 1,350, 1,380, 1,410, 1,440, 1,470] | 1,335 (90.8) |
| 2 | [1,380, 1,410, 1,440, 1,470, 1,500, 1,530, 1,560, 1,590, 1,620, 1,650] | 1,515 (90.8) |

### Model A

The datum Young's modulus ($E = 14$ GPa) was subtracted from the eight 1D Young's modulus continua (Fig. 2B), and the resulting difference continua were sign-permuted (Fig. 6A) to generate a number of artificial data samples. For each sample, the $t$ continuum was computed according to the typical one-sample $t$ statistic definition:

$$t(\boldsymbol{q}) = \frac{\overline{y(\boldsymbol{q})} - \mu(\boldsymbol{q})}{s(\boldsymbol{q})/\sqrt{N}} \tag{2}$$

where $\overline{y}$ is the sample mean, $\mu$ is the datum, $s$ is the sample standard deviation, $N$ is sample size and $\boldsymbol{q}$ is continuum position. Repeating for all permutation samples produced a distribution of 1D $t$ continua (Fig. 6B), whose maxima formed a 'primary' probability density function (PDF) (Fig. 6C). This primary PDF represents the expected maximum difference (from the datum case of $E = 14$ GPa) that smooth, purely random continua would be expected to produce if there were truly no effect.

We conducted classical hypothesis testing at $\alpha = 0.05$ using the primary PDF's 95th percentile ($t^*$) as the criterion for null hypothesis rejection; if the $t$ continuum associated with original, non-permuted data (Fig. 6A) exceeded $t^*$ the null hypothesis was rejected. In this example the original $t$ continuum failed to traverse $t^*$ (Fig. 6E) so the null hypothesis was not rejected. Based on the primary PDF the exact probability value was: $p = 0.101$ in the depicted example.

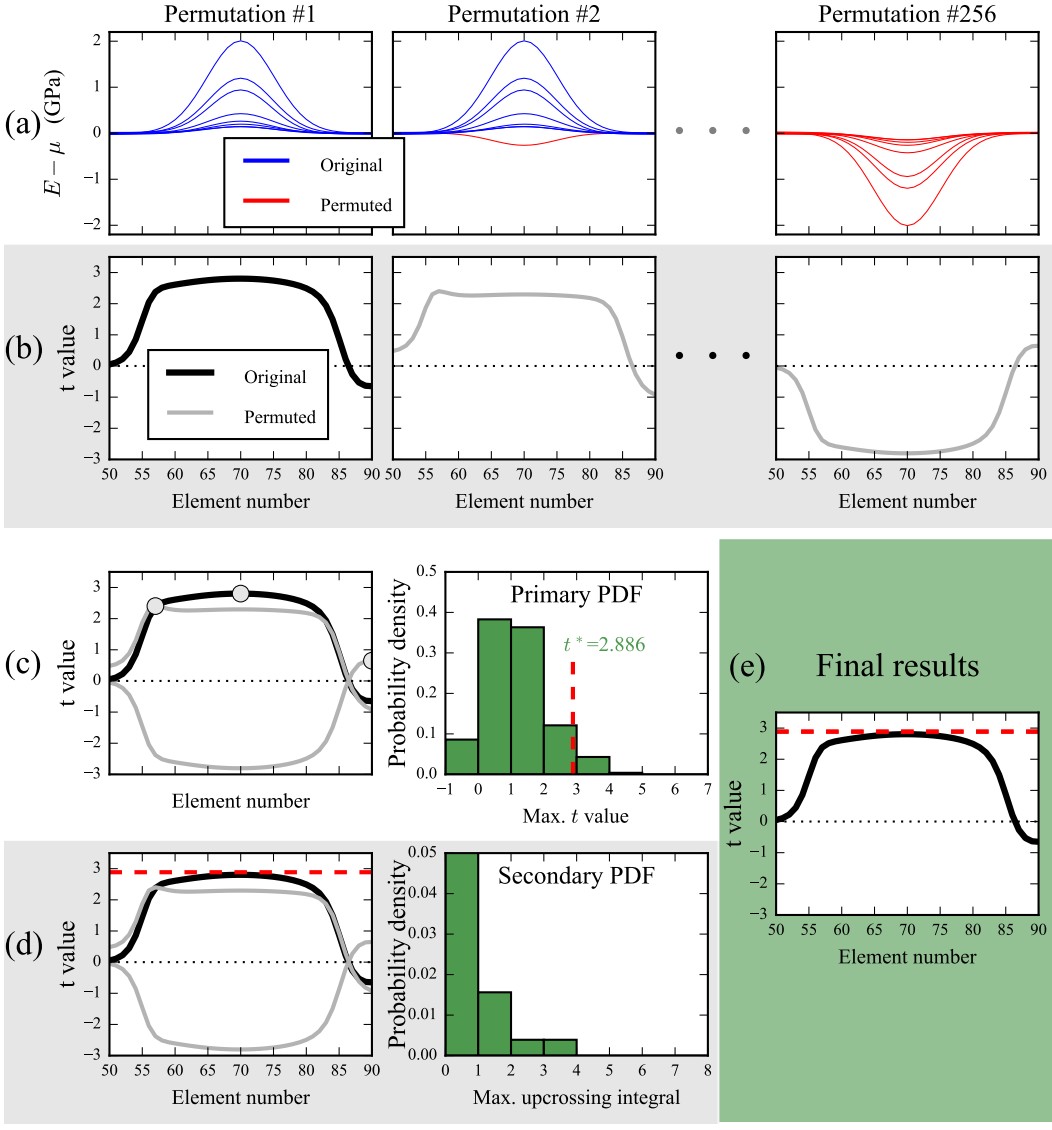

**Figure 6** **Depiction of non-parametric, permutation-based continuum-level hypothesis testing.** This example uses five of the Young's modulus continua from Fig. 2B and compares the mean continuum to the datum: $\mu = 14$ GPa. (A) Original continua were sign-permuted by iteratively multiplying subsets by $-1$. (B) For each permutation, a $t$ continuum was computed using Eq. (2). (C) The maximum $t$ values from all permutations were assembled to form a primary probability density function (PDF) from which a critical test statistic ($t^*$) was calculated. (D) Thresholding all permuted test statistic continua at $t^*$ produced upcrossings (Fig. 1) whose integral formed a secondary PDF from which upcrossing-specific $p$ values are computable. (E) Since the original test statistic continuum failed to traverse $t^*$ the null hypothesis was not rejected at $\alpha = 0.05$ for this example.

We repeated this procedure for the effective strain and von Mises stress distributions associated with the eight Young's modulus continua. In cases where the original $t$ continuum exceeded the $t^*$ threshold, probabilities associated with the upcrossing(s) (Fig. 1) were computed with a 'secondary' PDF (Fig. 6D) which embodied the probability of observing upcrossings with particular volume (i.e., supra-threshold integral). Note that

(i) $(1-\alpha)\%$ of the values in the secondary PDF are zero by definition, (ii) an upcrossing which infinitessimally exceeds $t^*$ has an integral of zero and a $p$ value of $\alpha$, and (iii) the minimum upcrossings $p$ value is $1/n$, where $n$ is the total number of permutations. All integrals were computed using trapezoidal approximation.

### Model A, part 2

We conducted a secondary analysis of Model A to examine how additional probabilistic variables increase computational demand. For this analysis we considered load direction $(\theta)$ to be uncertain, with a mean of zero and a standard deviation of 3 deg (forces with $\theta = 0$ deg are depicted in Fig. 2A, and these forces were rotated about the depicted $Y$ axis). For typical simulation of random variables hundreds or thousands of simulations are usually needed to achieve probability distribution convergence (*Dopico-González, New & Browne, 2009*), but we aimed to show that computational increases may be minimal for the proposed hypothesis testing framework.

We randomly varied $\theta$ for an additional 400 FE simulations, 50 for each of the observations depicted in Fig. 2B. We then qualitatively compared the permutation-generated distribution of $t$ continua after just 16 simulations (one extra FE simulation for each observation) to the distribution obtained after 400 FE simulations. To quantitatively assess the effects of the number of simulations $N$ on the distributions we examined the null hypothesis rejection rate for the $N = 16$ and $N = 400$ cases as a function of the number of post-simulation permutations.

### Model B

The goal of Model B analysis was to qualitatively assess the effects of imperfect contact geometry (Fig. 4) on both mean FE simulation results and statistical interpretations. Nine simulations were conducted for each of the three indenter faces (Fig. 4): one datum $(k = 820)$ and then the eight other values of $k$ as described above. For each indenter we computed the mean von Mises stress distribution in the compressed soft tissue, then compared this mean to the datum $(k = 820)$ stress distribution through the one-sample test statistic (Eq. 2).

### Model C

The goal of Model C analysis was to demonstrate how the analysis techniques and results for Model A and Model B extend to realistic, complex models. The null hypothesis of equivalent von Mises stress distributions in each group (Table 1) was tested using a slight modification of the permutation approach described above (Fig. 6). The only differences were that (i) the two-sample $t$ statistic was computed instead of the one-sample $t$ statistic, and (ii) group permutations were conducted instead of sign permutations. Group permutations were performed by randomly assigning each of the 20 continuum observations to one of the two groups, with ten observations in each group, then repeating for a total of 10,000 random permutations. Although the total number of possible permutations was $20!/(10! \; 10!) = 184{,}756$, we found no qualitative effect of adding more than 10,000 permutations.

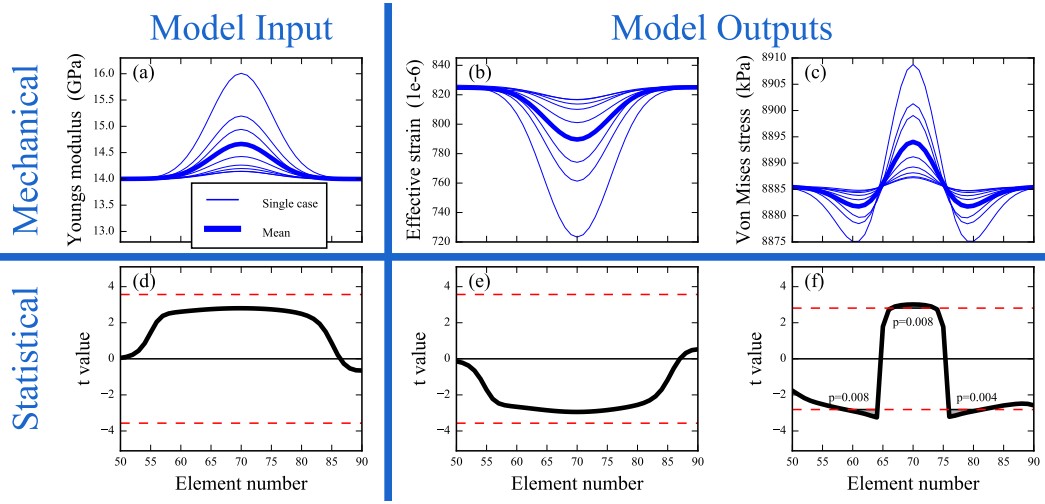

**Figure 7  Model A results.** (A–C) Young's modulus input observations and strain/stress continua associated with each observation. (D–F) Hypothesis testing results ($\alpha = 0.05$); red dotted lines depict critical thresholds.

## RESULTS

### Model A

FE simulations of each of the eight cases depicted in Fig. 2B yielded the stress/strain distributions and $t$ statistic distributions depicted in Fig. 7. In this example Young's moduli only increased (Fig. 7A) and strain only decreased (Fig. 7B), but stress exhibited central increases (near element #70) and peripheral decreases (near elements #60 and #80) (Fig. 7C), emphasizing the nonlinear relation between model inputs and outputs.

Maximum absolute $t$ values differed amongst the field variables (Figs. 7D–7F), with stress exhibiting the largest maximum absolute $t$ values. The null hypothesis was rejected for von Mises stresses but not for either Young's modulus or effective strain. Additionally, both stress increases and stress decreases were statistically significant (Fig. 7F). These results indicate that statistical signal associated with the Young's modulus inputs was amplified in the von Mises stress field, but we note that strain would have been the amplified variable had the the model been displacement–loaded instead of force-loaded. More generally, these results show that statistical conclusions pertaining to different model variables can be quite different, and that different continuum regions can respond in opposite ways to probabilistic inputs.

Although stiffness increased non-uniformly as a Gaussian pulse (Fig. 7A) the test statistic magnitude was effectively uniform across that region (elements 60–80; Fig. 7D). This suggests that mechanical and statistical magnitudes are not directly related, and thus that statistical conclusions mustn't be limited to areas of large mechanical signal unless one's hypothesis pertains specifically to those areas.

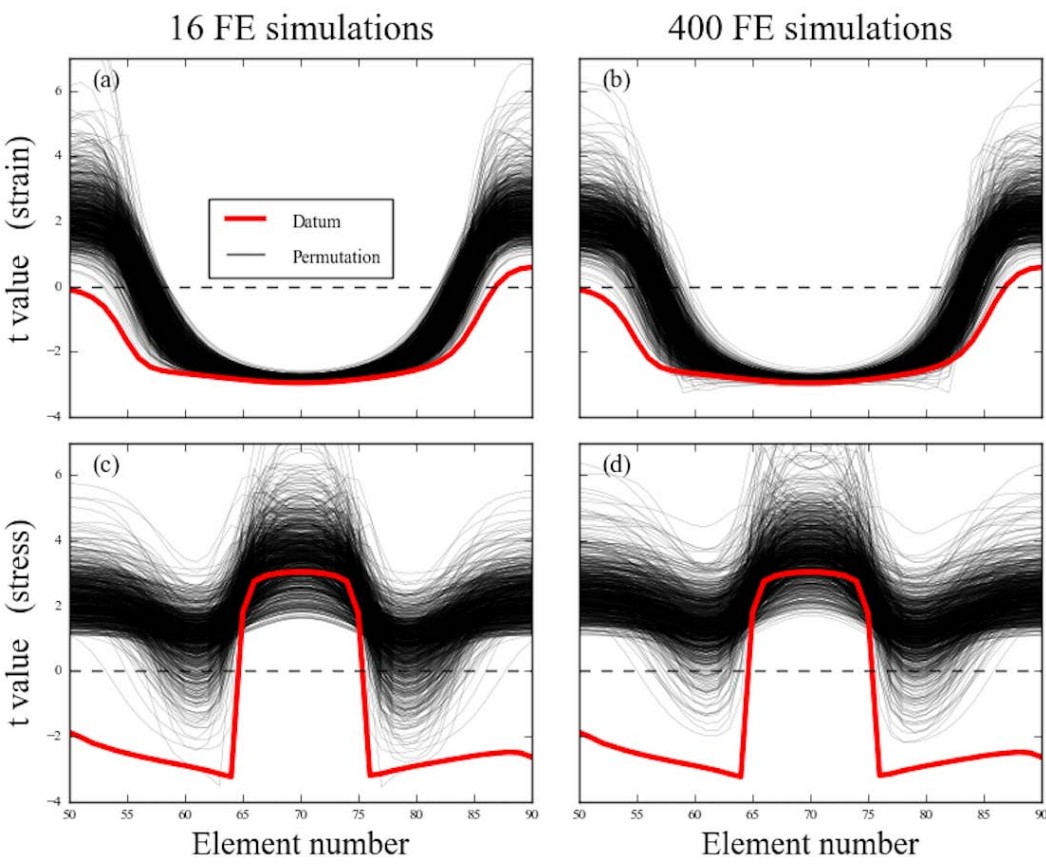

**Figure 8** **Model A uncertainty results.** Strain (A, B) and stress (C, D) under a load direction uncertainty with a standard deviation of 3 deg.

## Model A, part 2

Adding uncertainty to the load direction increased variability and thus caused absolute $t$ value decreases near element #70 (Fig. 8A), but general loading environment changes caused increases to absolute $t$ values in other model areas, especially toward elements #50 and #90. The stress response was somewhat different , with absolute $t$ values increasing near element #70 but decreasing elsewhere (Fig. 8C), re-emphasizing the complex relation amongst different field variables' response to probabilistic model features.

The $t$ distributions for stress and strain were not qualitatively affected by the number of additional FE simulations; 16 simulations, or one extra simulation per observation (Figs. 8A and 8C) yielded essentially the same results as 400 simulations (Figs. 8B and 8D). The reason is that permutation leverages variability in small samples to produce a large number of artificial samples, and thereby approximates the results of a large number of FE simulations.

To quantify $t$ continuum distribution stability as a function of the number of permutations we considered the null hypothesis rejection rate in both cases of 16 and 400 FE simulations (Fig. 9). After approximately 200 permutation iterations the null hypothesis rejection rate was effectively identical for both 16 and 400 FE simulations.

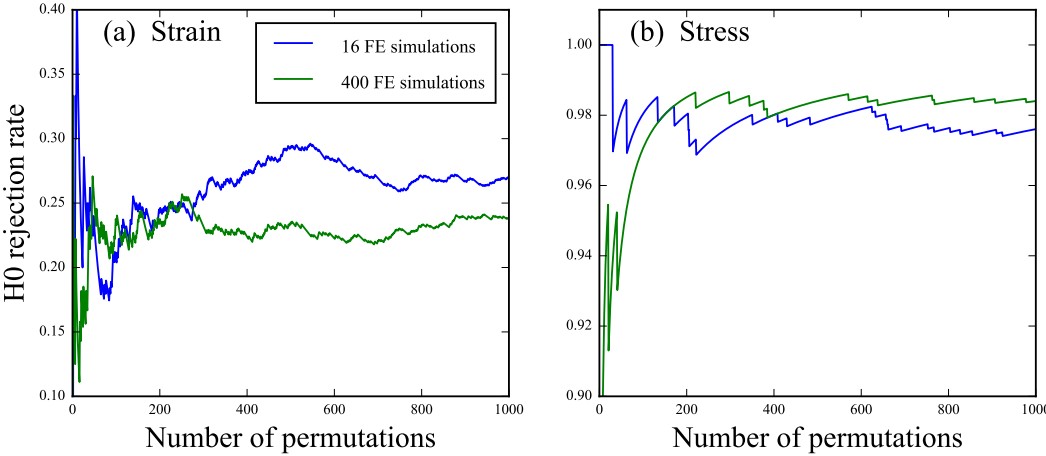

**Figure 9** **Model A convergence results.** Null hypothesis (H0) rejection rate as a function of the number of permutations for both 16 and 400 FE simulations.

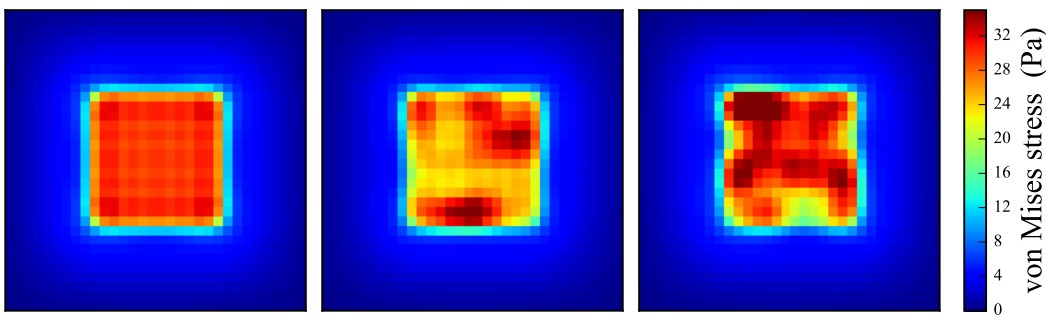

**Figure 10** **Model B results.** Mean stress distributions for the three indenter faces. Note that these patterns closely follow the indenter face geometry depicted in Fig. 4.

These results suggest that permutation, which is extremely fast compared to FE simulation, may be able to effectively approximate a large number of FE simulations using the results of only a few FE simulations.

## Model B

The mean stress distributions associated with the three indenter faces (Fig. 10) closely followed indenter face geometry (Fig. 3). Variation in material parameters was associated with stress distribution variability (Fig. 11A). Nevertheless, $t$ values were effectively constant across all elements and all three models (Fig. 11B). This suggests that test statistic continua are more robust to model geometry imperfections than are stress/strain continua.

## Model C

A two-sample $t$ test regarding the material parameters (Table 1) yielded $t = 5.17$, $p < 0.001$ and thus a rejection of the null hypothesis of equal group means. These probabilistic material parameters produced mean stresses which were generally higher in Group B vs. Group A (Fig. 12), where a stress distribution difference plot clarified that inter-group

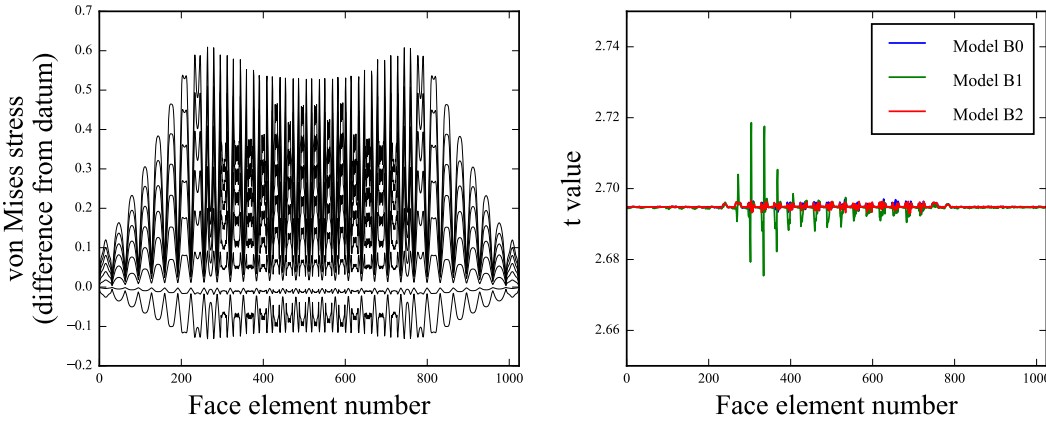

**Figure 11  Model B random simulation results.** (A) Large variation was present amongst the individual continua (only Model B0 results shown). (B) Test statistic continua were effectively constant in all elements and across all three indenters.

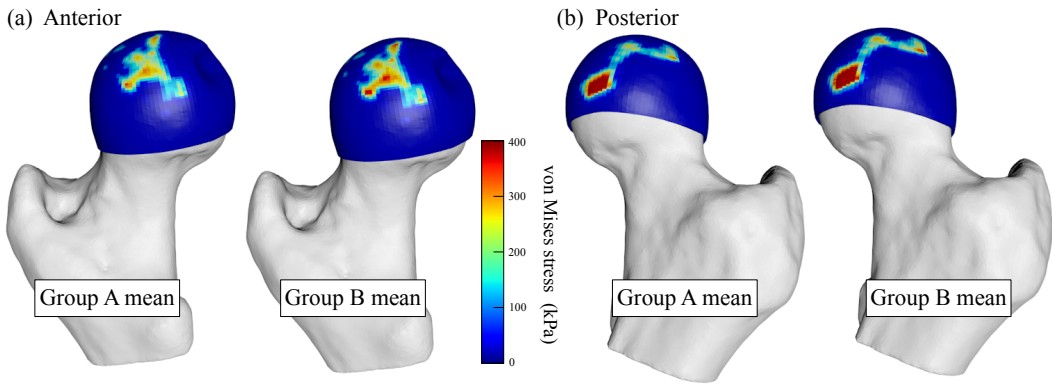

**Figure 12  Model C results.** Mean stress distributions.

differences were generally confined to areas of large stress (Fig. 13). The inter-group statistical differences were much broader, covering essentially the entire femoral cartilage (Fig. 14). Moreover, relatively broad regions of the cartilage exhibited significant stress *decreases*, similar to the result observed in the simple bone model (Fig. 7F).

These results reiterate many of the aforementioned methodological points. In particular, changes in probabilistic model inputs (in this case: material parameter values) can have statistical effects on output fields (in this case: von Mises stresses) which are not easily predicted. Additionally, the visual advantages of full-field analyses are somewhat clearer in this more anatomically correct model; tabulated stresses from different regions of the femoral cartilage would be more difficult to interpret in terms of the original anatomy. Last, mechanical (Fig. 13) and statistical (Fig. 14) results can be quite different.

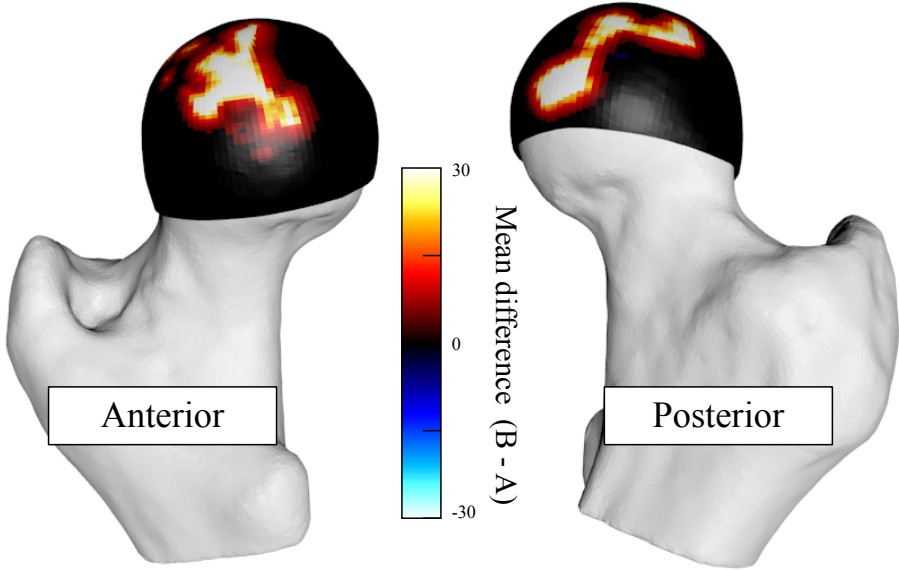

**Figure 13  Model C inter-group results.** Mean stress difference.

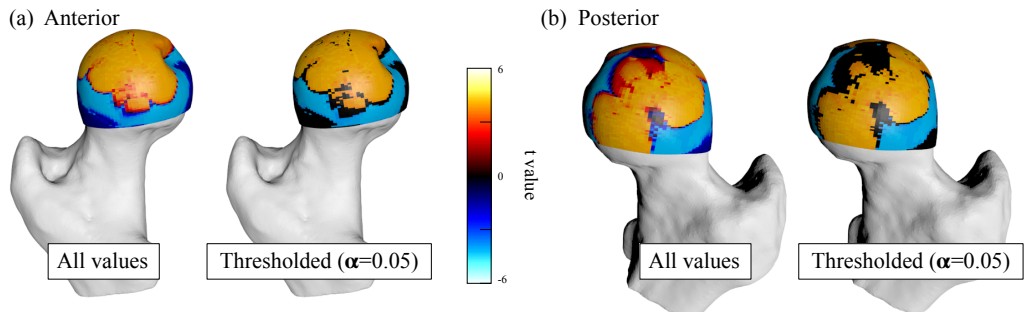

**Figure 14  Model C inter-group statistical results.** Raw and thresholded *t* statistic distributions.

## DISCUSSION

This paper demonstrated how a non-parametric permutation technique from Neuroimaging (*Nichols & Holmes, 2002*) can be used to conduct classical continuum-level hypothesis testing for finite element (FE) models. It's main advantages are:

1. Easy implementation. As demonstrated in this project's software repository (http://github.com/0todd0000/probFEApy), non-parametric hypothesis testing for FE models can be implemented using relatively compact scripts.

2. Computational efficiency. After simulating subject-specific results—which is usually necessary in arbitrary multi-subject studies—no additional FE simulations are needed; permutation can operate on pre-simulated small-sample results to approximate large-sample probabilities (Fig. 6). Producing the main Model A results (Fig. 7) required a total of only 1.3 s to execute on a desktop PC, including both FE simulations and permutation-based probability computation.

3. Non-measured uncertainty capabilities. Adding uncertainty in the form of random model parameters does not necessarily require large increases in computational demands; results suggest that with respect to an original dataset with $N$ simulations, it may be possible to robustly accommodate additional uncertainty with just $N$ additional simulations (Figs. 8–9).

4. Visual richness and tabulation elimination. Continuum-level hypothesis testing results can be presented in the same geometric context as commonly visualized field variables like stress and strain (Figs. 7B, 7E and Figs. 7C, 7F), which eliminates the need to separately tabulate statistical results.

5. Arbitrarily complex experiments. While only one- and two-sample designs were considered here, $t$ statistic continua generalize to $F$ and all other test statistic continua, so arbitrarily complex designs ranging from regression to MANCOVA can be easily implemented using permutation.

6. Robustness to geometric imperfections. Small geometric changes can have qualitatively large effects on stress/strain continua, but have comparably little-to-no effect on test statistic continua (Fig. 8), implying that continuum-level hypothesis testing may be more robust than commonly employed procedures which analyze local maxima. This potential danger is highlighted in the more realistic Model C, whose mean differences (Fig. 13) exhibited high focal stresses whereas the statistical continuum was much more constant across the contact surface (Fig. 14).

## Mechanical vs. statistical interpretations

Mechanical and statistical continua are generally different. For example, for Model A it is clear that each stiffness increase (Fig. 2B) has *mechanical* effects on the strain/stress continuum, but the *statistical* effects are less clear because there is relatively large uncertainty regarding the true nature of the stiffness increase in the population that this sample represents. For classical hypothesis testing, mechanical meaning is irrelevant because all mechanical effects must be considered with respect to their uncertainty. Further emphasizing the tenuous relation between mechanical and statistical meaning are regions of small mechanical signals (for Model A: near the periphery of the stiffness increase region) which can be accompanied by relatively large statistical signals.

To objectively conduct classical hypothesis tests on FEA results, it is therefore essential to explicitly identify the hypothesis prior to conducting simulations. If limiting analyses to only areas of large mechanical signal can be justified in an *a priori* sense, then those, and only those areas should be analyzed without any theoretical problem. If, however, one's *a priori* hypothesis pertains to general stress/strain distribution changes, and not specifically to areas of high mechanical signal, it may be necessary to consider the entire model because maximal mechanical and maximal statistical signals do not necessarily coincide.

## Comparison with common techniques

In the literature, FE-based classical hypothesis testing is typically conducted via scalar analysis of local extrema (*Radcliffe & Taylor, 2007*). Applying that approach to the local mechanical change extrema in Model A (Figs. 7A–7C) yielded the results in Table 2. The

**Table 2  Model A results.** Analyses of local extrema (at element 70) using a non-parametric permutation-based two-sample $t$ test. SD, standard deviation.

| Variable | Mean | SD | $t$ | $p$ |
|---|---|---|---|---|
| Young's modulus (GPa) | 14.665 | 0.670 | 2.804 | 0.026 |
| Effective strain (1e−6) | 789.6 | 33.9 | −2.946 | 0.022 |
| von Mises stress (kPa) | 8894.0 | 8.0 | 3.014 | 0.020 |

null hypothesis (of no mean change with respect to the 14 GPa case) was rejected at $\alpha = 0.05$ for all three mechanical variables: Young's modulus, effective strain and von Mises stress.

While the test statistic magnitudes are the same for both the proposed whole-model approach (Fig. 7) and these local extremum analyses, the critical threshold at $\alpha = 0.05$ is different because the spatial scope is different. The broader the spatial scope of the hypothesis, the higher the threshold must be to avoid false positives (Friston et al., 2007); in other words, random processes operating in a larger volume have a greater chance of reaching an arbitrary threshold.

The proposed model-wide approach (Fig. 7) and the local extremum (scalar) approach have yielded contradictory hypothesis testing conclusions for both Young's modulus and strain distributions, so which approach is correct? The answer is that both are correct, but both cannot be simultaneously correct. The correct solution depends on the *a priori* hypothesis, and in particular the spatial scope of that hypothesis. If the hypothesis pertains to only the local extremum, then the local extremum approach is correct, and whole-model results should be ignored because they are irrelevant to the hypothesis. Similarly, if the hypothesis pertains to the whole model, then the whole model results are correct and local extrema results should be ignored because they are irrelevant to the hypothesis. We would argue that all FE analyses implicitly pertain to the whole model unless otherwise specified, and that focus on specific scalar metrics is appropriate only if justified in an *a priori* manner.

Historically in biomechanical FEA, low sample sizes (frequently $n = 1$ for each model) permitted nothing more than qualitative comparisons of stress or strain maps, and/or numerical comparison of output parameters at single nodes. Nevertheless conventional FEA can concurrently and ironically suffer from an excess of data when results are tabulated over many regions, often in a non-standardized manner across studies.

With the continued increase of computer power and processing speed, FE models comprising over one million elements are becoming more and more common (e.g., *Moreno et al., 2008*; *Bright & Rayfield, 2011a*; *Cox, Kirkham & Herrel, 2013*; *Cox, Rinderknecht & Blanco, 2015*; *Cuff, Bright & Rayfield, 2015*). Yet, typically stress and strain values are only reported and analysed from just a few elements (*Porro et al., 2013*; *Fitton et al., 2012a*). Alternatively, average or peak stress or strain values can be computed for whole models (*Dumont et al., 2011*; *Cox et al., 2012*; *Parr et al., 2013*; *Sharp & Rich, in press*) or selected regions (*Wroe et al., 2007a*; *Wroe et al., 2007b*; *Nakashige, Smith & Strait, 2011*). The recent application of geometric morphometrics to FEA results (*Cox et al., 2011*; *Fitton et al., 2012b*; *O'Higgins & Milne, 2013*) has gone some way to providing a method of analysing whole models rather than individual elements, but is limited to the analysis of deformations.

The approach outlined here enables, for the first time, the analysis of all stresses or strains in a single hypothesis test.

Another major benefit of the technique outlined here is its ability to take in consideration input parameters that are only imprecisely known. When modelling biological structures, the material properties of the model, and the magnitude and orientations of the muscle loads cannot always be directly measured. This is an especially acute problem in studies dealing with palaeontological taxa. Previous research has addressed this issue principally by the use of sensitivity analyses which test the sensitivity of a model to changes in one or more unknown parameters (*Kupczik et al., 2007*; *Bright & Rayfield, 2011a*; *Cox et al., 2011*; *Cox, Rinderknecht & Blanco, 2015*; *Reed et al., 2011*; *Wood et al., 2016*; *Toro-Ibacache et al., 2016*). The models are identical save for the unknown parameters, which are then varied between extremes representing likely biological limits or the degree of uncertainty. In such studies, the number of different models is usually quite low, with each parameter only being tested at a maximum of five different values. Our method takes this approach to its perhaps logical extreme—the unknown parameter is allowed to vary randomly within defined limits over a large number of iterations (usually on the order of 10,000). These iterations produce a distribution of results that can be statistically compared with other such distributions.

A final advantage is that statistical continua may be less sensitive to geometric mesh peculiarities than stress/strain continua. In Fig. 10 and Fig. 13, for example, it is clear from the oddly shaped regions of stress difference that these effects were likely caused by mesh irregularities and that remeshing would likely smooth out these areas of highly localized stress changes. The test statistic continuum, on the other hand, appeared to be considerably less sensitive to localization effects (Fig. 11) and (Fig. 14). This may imply that one needn't necessarily develop an ideal mesh, because statistical analysis may be able to mitigate mesh peculiarity-induced stress distribution irregularities.

## Limitations

The major limitation of the proposed method as it currently stands is that only models of identical geometry can be compared. Thus, while the technique can be readily used to address sensitivity-like questions regarding material properties, boundary conditions and orientations, the method cannot readily address geometry-relevant questions, such as are created by varying mesh density (*Bright & Rayfield, 2011b*; *Toro-Ibacache et al., 2016*), or are found in between-taxa analyses (*Dumont, Piccirillo & Grosse, 2005*; *Dumont et al., 2011*; *Oldfield et al., 2012*; *Cox et al., 2012*; *Wroe et al., 2007a*; *Sharp, 2015*). Nevertheless, through three-dimensional anatomical registration (*Friston et al., 2007*) and also potentially intra-model spatial interpolation to common continuum positions $q$ (Eq. 2), it may be possible to apply the technique to arbitrary geometries even in cases of large deformation and/or geometrical disparity (*Schnabel et al., 2003*).

A second limitation is computational feasibility. Although our results suggest that incorporating a single additional uncertain parameter into the model may not greatly increase computational demand, this may not be true for higher dimensional parameter spaces. In particular, given $N$ experimental measurements, our results show that $2N$

simulations are sufficient to achieve probabilistic convergence (Fig. 9). However, this result may be limited to cases where the uncertainty is sufficiently small so that it fails to produce large qualitative changes in the underlying stress/strain continua. Moreover, the feasibility for higher-dimensional parameter spaces is unclear. In particular, a sample of $N$ observations is likely unsuitable for an $N$-dimensional parameter space, or even an $N/2$-dimensional parameter space. The relation between uncertainty magnitude, number of uncertain parameters, the sample size and the minimum number of FE simulations required to achieve probabilistic convergence is an important topic that we leave for future work.

A third potential limitation is that both upcrossing features and the test statistic continuum can be arbitrary. In this paper we restricted analyses to the upcrossing maximum and integral due to the robustness of these metrics with respect to other geometric features (*Zhang, Nichols & Johnson, 2009*). Other upcrossing metrics and even arbitrary test statistic continua could be submitted to a non-parametric permutation routine. This is partly advantageous because arbitrary smoothing can be applied to the continuum data, and in particular to continuum variance (*Nichols & Holmes, 2002*), but it is also partly a disadvantage because it increases the scope of analytical possibilities and thus may require clear justification and/or sensitivity analyses for particular test statistic and upcrossing metric choices.

A final limitation is that the both the test statistic and probability continua are directly dependent on the uncertainty one selects via model parameter variance. This affords scientific abuse because it allows one to tweak variance parameters until the probabilistic results support one's preferred interpretation. We therefore recommend that investigators both clearly justify variance choices and treat variance itself as a target of sensitivity analysis.

## Summary

This paper has proposed a probabilistic finite element simulation method for conducting classical hypothesis testing at the continuum level. The technique leverages probability densities regarding geometric features of continuum upcrossings, which can be rapidly and non-parametrically estimated using iterative permutation of pre-simulated stress/strain continua. The method yields test statistic continua which are visually rich, which may eliminate the need for tabulated statistical results, which may reveal unique biomechanical information, and which also may be more robust to mesh and other geometrical model peculiarities than stress/strain continua.

### Funding

This work was supported in part by an International Exchanges Scheme grant from the Royal Society (UK) and Wakate A grant 15H05360 from the Japanese Society for the Promotion of Science. The funders had no role in study design, data collection and analysis, decision to publish, or preparation of the manuscript.

### Grant Disclosures

The following grant information was disclosed by the authors:

International Exchanges Scheme.
Wakate A: 15H05360.

## Competing Interests

Philip G. Cox is an Academic Editor for PeerJ.

## Author Contributions

- Todd C. Pataky conceived and designed the experiments, performed the experiments, analyzed the data, wrote the paper, prepared figures and/or tables, performed the computation work, reviewed drafts of the paper.
- Michihiko Koseki and Phillip G. Cox conceived and designed the experiments, analyzed the data, wrote the paper, reviewed drafts of the paper.

## Data Availability

Raw data (including models and associated scripts) are available at:
https://github.com/0todd0000/probFEApy.
The entire repository can be direct-downloaded from:
https://github.com/0todd0000/probFEApy/archive/master.zip.

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
