# Peer review of "Probabilistic biomechanical finite element simulations: whole-model classical hypothesis testing based on upcrossing geometry"

_PeerJ Computer Science, doi:10.7717/peerj-cs.96_

## Round 0.1 · original submission · Major Revisions

Please address all of the referees' comments in your revision.

·

Basic reporting

No comments.

Experimental design

No comments.

Validity of the findings

No comments.

Additional comments

In this manuscript, the authors describe a strategy to understand the statistical implications of model input variations/uncertainty on model outputs obtained through the whole geometry. By calculating statistical parameters throughout the whole model domain, representing them as a field variable and proposing a non-parametric analysis to identify geometric regions that may have significance, the authors provide the possibility to move beyond the abridged analysis of simulation results based on scalar outcomes, e.g. peak von Mises stresses. The manuscript is well written, the concepts are conveyed with good examples, and the limitations of the work were discussed in a detailed manner. This reviewer is in favor of publication of this study. Yet, a few issues are noted below:

- The Mooney-Rivlin formulation noted in the manuscript, Eqn. 7, does not seem to match the one used in FEBio, particularly when volumetric component is concerned. The authors may want to check FEBio theory manual to confirm.

- It is not clear how non-parametric permutation-based probabilities are calculated for model outputs, i.e., using "sign permutation" directly on model outputs OR conducting "sign permutation" on model inputs and then conducting finite element analysis. Clarification of this is important as it dictates the required number of potentially costly simulations.

- In section 5.4, the authors note an important limitation - the methodology being suitable to designs involving only a small number of unknown parameters. While this is understandable, it may prevent conducting useful studies. It will be useful to know more about at what level such analysis become not feasible. For example, is it possible to use Model A, this time have a failure threshold parameter (variable from model to model and from element to element), where von Mises stress can be compared against to see where in the geometry failure will likely happen. It may be exciting to provide such a study as an example with two model inputs with uncertainties.

- The first author has done similar work on temporal analysis of 1D signals. How does this study relates to the author's previous work? Is it an extension to field problems, yet still 1D in terms on model input?

- It will be beneficial to disseminate the Python scripts, at least for Model A, for the readers to be able to reproduce the results and play with the capabilities of the described method.

- The authors provide a useful comparison in the last paragraph of section 5.2. Nonetheless, this paragraph may need to strengthened with examples to guide the reader about the situations where local extremum approach may be more appropriate than the model-wide approach and vice versa. It is easy to be a proponent of local extremum approach simply because bad things likely happen at locations of highest stresses, e.g. failure. The example on local extremum on mesh sensitivity is appreciated. Yet, this reviewer wonders at what type of physiological situations knowledge about non-local extremum model regions become useful. For the sake of argument (and realizing that Model C was not necessarily provided for physiological investigations), does it really matter to know that strains in ligament attachment sites are highly sensitive to internal/external rotation when they are not necessarily deforming that much to begin with?

- Please correct the phrase ".. there is a relatively large uncertainty amount of uncertainty .."

Reviewer 2 ·

Basic reporting

The paper is well written and complies with the PeerJ policies and formats. However, it is hard to put it in perspective with the current practices in probabilistic finite element methods, and the practical advantages and implications of the proposed approach. Please consider the following questions/comments to further improve your manuscript:

1. Uncertainty quantification in computational methods is nowadays a mature field of research and there exist many competing methods (e.g., polynomial chaos, ANOVA, non-parametric surrogates based on Gaussian processes, etc.) for analyzing continuum statistics and propagating probability densities through nonlinear systems using FE simulations, experiments, etc. Such techniques are never mentioned in the manuscript, and it is not clear how the proposed method connects and compares against those well-established approaches. In particular, the methodology based on t-statistics outlined in Sec. 2.2 seems quite simplistic, and the conclusions drawn in lines 134-144 and section 5.1 are quite obvious. Please, provide a detailed discussion on the motivation and merits of the proposed approach as compared with the current-state-of-the-art in uncertainty quantification.

2. p.3, l.93-94: Is it this the noise that causes the asymmetry observed in the t-statistic shown in Fig.3b? What is the purpose of adding this small random signal?

3. p.4, l. 113: The abbreviation “SD” is introduced without explanation. Please spell out every abbreviation the first time it is used in the text.

4. Please provide sufficient information so that the reader can reproduce every numerical example presented. For example, provide a brief description of which equations are solve using FE? What are the boundary/initial conditions, etc?

5. p.7, l.196-197: how do you select the nodes? does this matter a lot? do you use quadrature to compute the upcrossing integral? if yes, is the selection of the nodes and the quadrature related?

6. Figure 6: Figure captions should should be self contained. Please briefly explain what we see in a,b,c,d.

7. p.14, l.386-388: Please provide a more detailed comment on the difficulties introduced by high- dimensional parameter spaces. How does the computational cost and feasibility of the proposed approach scale in such cases?

Experimental design

The submission describes original research and is in accordance with the scope of the journal. Regarding result reproducibility please see item 4.) in the Basic Reporting section.

Validity of the findings

The submission meets the scientific standards of the journal. One minor concern is that there seems to be no connection with the current literature on uncertainty quantification in computational science. Moreover, the technical approach is rather simplistic and the conclusions drawn may seem obvious to a reader acclimated with probabilistic finite element simulations.

---

## Round 0.2 · accepted · Accept

Thank you for your revision submission, and for your careful
attention to the referee comments.